# The Dielectric Properties Improvement of Cable Insulation Layer by Different Morphology Nanoparticles Doping into LDPE

**Guang Yu [1], Yujia Cheng [1,\*] and Xiaohong Zhang [2]**

[1]  Mechanical and Electrical Engineering Institute, University of Electronic Science and Technology of China, Zhongshan Institute, Zhongshan 528400, China; yuguang@hrbust.edu.cn

[2]  Key Laboratory of Engineering Dielectrics and Its Application, Ministry of Education, Harbin University of Science and Technology, Harbin 150080, China; x_hzhang2002@hrbust.edu.cn

\*  Correspondence: chengyujia@hrbust.edu.cn; Tel./Fax: +86-451-8639-0800

**Abstract:** Low density polyethylene (LDPE) doped with inorganic nano-MMT and nano-ZnO particles improved the dielectric properties of the cable insulation layer. In this article, nano-MMT/LDPE and nano-ZnO/LDPE composites were prepared by polymer intercalation and melt blending, respectively. The octadecyl quaternary ammonium salt and silane coupling agent were applied for surface modification in nano-MMT and nano-ZnO particles, and this then improved the compatibility of nanoparticles and polymeric matrix. These samples were characterized by FTIR, PLM, DSC and TSC, from which the effect of nanoparticles doping on polymer crystal habit and interface traps would be explored. In these experiments, the AC breakdown characteristics and space charge characteristic of different composites were studied. The experimental results showed that the interface bonding of nanoparticles and polymer was improved by coupling agents modifying. The dispersion of nanoparticles in matrix was better. When the mass fraction of nanoparticles doping was 3 wt.%, the crystallization rate and crystallinity of composites increased, and the crystalline structure was more complete. Besides, the amorphous regions in material decreased and the conducting channel was circuitous. At this time, the breakdown field strength of nano-MMT/LDPE and nano-ZnO/LDPE increased by 10.3% and 11.1%, compared to that of pure LDPE, respectively. Furthermore, the density and depth of interface traps in polymer increased with nanoparticles doping. Nano-MMT and nano-ZnO could both restrain the space charge accumulation, and the inhibiting effect of nano-ZnO was more visible.

**Keywords:** different crystalline morphology; dielectric properties; insulation layer; nanoparticles; LDPE

## 1. Introduction

With the rapid development of electric power industry all over the world, energy optimization and reasonable allocation were the most essential problems to be solved at present. Polyethylene (PE) as a kind of nonpolar high molecular polymer, because of its excellent characteristics such as high insulation resistance, low dielectric constant, low loss and being hardly affected by temperature, was widely used in substrate materials of the cable insulation layer. Due to the influence of various factors such as electrical, heat and mechanical, the insulating property of cable material would decrease during the long term operation. Numerous studies also showed that these factors were critical for the stable operation and service life of cables [1–5]. Therefore, the cable main insulation material modification through various ways was an important strategy to improve the combination property of insulation material.

Nanoparticles had many excellent chemical and physical properties, such as surface effect, quantum dimension effect, small size effect and macro quantum effect [6–8]. The synergistic reaction of nanoparticles and polymer matrix made the polymer nanocomposites possess unique mechanical properties [9–12]. Related experimental results demonstrated that nanoparticles doping effectively restrained the charges accumulation in the polyethylene matrix [13–15]. At the same time, the volume resistivity and breakdown field strength increased [16,17]. Nano-MgO and nano-ZnO doping had effective functions to improve the electrical properties of polymer composites, such as partial discharge-resistance, electrical treeing, corona aging and space charge accumulation restriction. In the 1980s, the conception of the nanocomposite was submitted by Roy [18,19]. Unlike single composite material, the nanocomposite possessed plenty of unique properties [20,21]. In recent years, as the most important influential factors, the compatibility and interface state of polymer with inorganic particles had been widely and deeply studied. But the effect of different morphologies' nanoparticles on composite interfacial structure, microdomain structure and crystal structure had not been well-studied. Therefore, the layered nano-MMT and spherical nano-ZnO, respectively, were doped in polyethylene in this article. From chemical structure test and crystalline structure test, the effect of different morphologies' nanoparticles on interface traps structure and crystalline morphology in polymer was studied. From the breakdown characteristic test and space charge characteristic test, the effect of nano-MMT and nano-ZnO doping on the electrical property of polyethylene was explored. Combining microcosmic test and macroscopic experimental research, it was known that the nanoparticles doping improved the dielectric properties of the material.

## 2. Experiment Materials and Nanoparticles Modification

### 2.1. Raw Materials and Equipments

Raw materials and equipment of nanocomposite preparation in this article were: Low density polyethylene (LDPE, DaQing Petrochemical, Daqing, China), of which the density distribution was 0.910–0.925 mg/cm$^3$, nano-MMT with a cation exchange capacity of 80 mol/100 g, nano-ZnO (DK nano) with a particle size of 30 nm, silane coupling agent (KH-570, Zhejiang Feidian Chemical Co., Ltd., Zhejiang, China), octadecyl trimethyl ammonium chloride, 1010 antioxidant, ultrasonic concussion cleaner (JP-120ST, JieMeng Technologies Co., Ltd., Hangzhou, China), rotating speed rheometer (CTR-100, Hapro Electric Technology Co., Ltd., Harbin, China), press vulcanizer (XL8, DongGuan Bolon Precision Testing Machines, Dongguan, China), polarizing microscope (MXP6000-X4, Eiscope Optical Technology, Guangzhou, China), differential scanning calorimeter (DSC214, NETZSCH Group, Lanzhou, China) and Fourier-transform infrared spectroscopy (FTIR, IR-2000, JingTuo Science and Technology Ltd., Tianjin, China). The main equipment were shown in Figure 1.

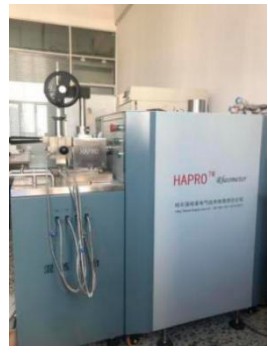 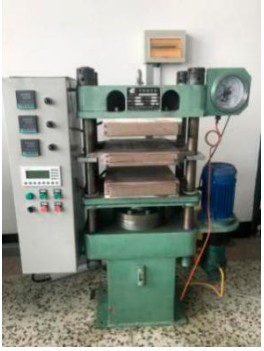 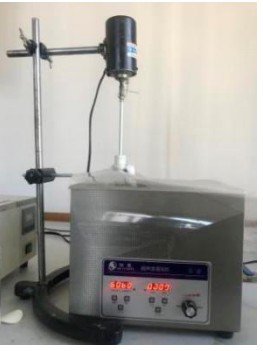 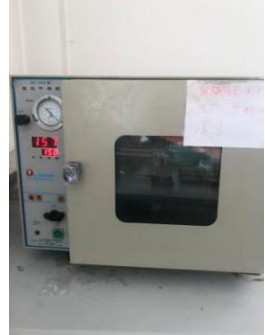

(**a**) Rheometer     (**b**) Press vulcanizer     (**c**) Drying oven     (**d**) Ultrasonic cleaner

**Figure 1.** Main experimental equipment.

## 2.2. Surface Modification of Different Nanoparticles

Nano-MMT particles surface modification was processed by octadecyl trimethyl ammonium chloride. Firstly, the moderate unmodified MMT (N-MMT) and deionized water were added into a flask with three necks. After a period of mixing, the suspension appeared. Secondly, the moderate octadecyl trimethyl ammonium chloride was weighed, based on the cation exchange capacity of N-MMT. With the ionized water mixing, it would form a solution. This solution was added into the suspension of N-MMT drop by drop. The reactants appeared with 2 h stirring in 80 °C, then the deionized water was used to wash and filtrate them. The 1% silver nitrate solution was dropped into a reactant until the white precipitate disappeared. Finally, the modified MMT (O-MMT) was prepared by drying, grinding and sifting [22]. The ion exchange process of MMT was shown in Figure 2.

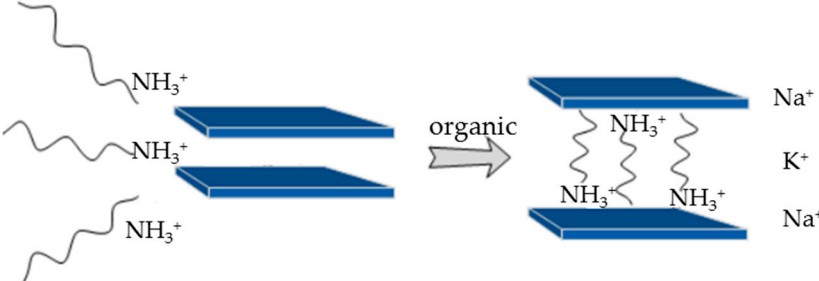

**Figure 2.** Diagram of the alkyl ammonium salt's modification effects on MMT.

Nano-ZnO particles surface modification was processed by KH-570 silane coupling agent. Firstly, the samples of unmodified ZnO (N-ZnO) was integrated into the mixture of absolute alcohol and distilled water. Secondly, the solution was dealt with ultrasonic oscillation for 1 h, and stirring in 80 °C for 2 h. After that, the silane coupling agent was added into the solution slowly. The suction flask was used for decompression, and a Büchner funnel was used to filter the mixture. Finally, the modified ZnO (O-ZnO) was prepared by drying, grinding and sifting. The KH-570 silane coupling agent was a special kind of surface modification agent, which possessed organic and inorganic groups simultaneously. The hydrogen bond could be formed by the dehydration condensation reaction of hydroxyls and hydrogen ions on the surface [23]. It would introduce a number of organic groups in nano-ZnO particles surface, and the nanoparticles then better combined with the polymeric matrix. The surface modification mechanism of the silane coupling agent was shown in Figure 3.

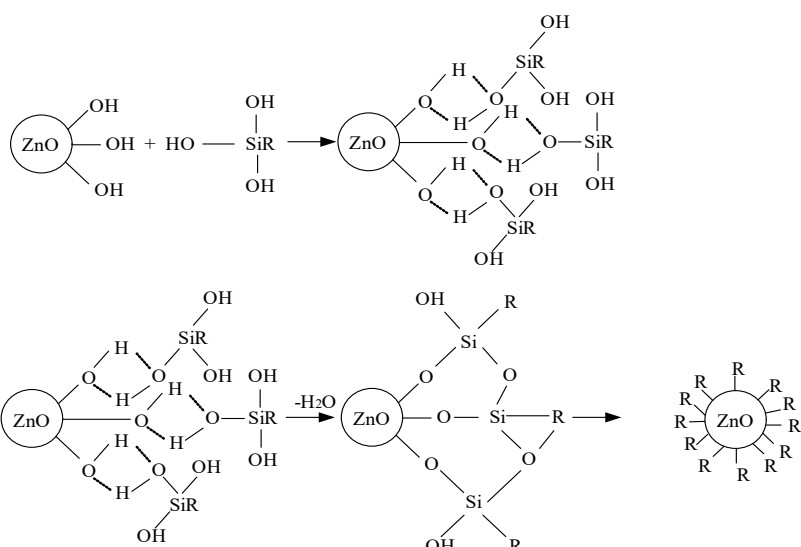

**Figure 3.** Surface modification of nano-ZnO with silane coupling agent.

### 2.3. Nanocomposites Preparation

In this article, nano-MMT/LDPE and nano-ZnO/LDPE composites were prepared by melt intercalation and melt blending, respectively. According to the related research results [24], the polymer doped with moderate inorganic nanoparticles improved the dielectric property of the matrix effectively. However, too much of the doping amount would lead to agglomeration, while too little would lead to the nanometer effect decrease. Therefore, the doping amount of nano-MMT and nano-ZnO was 3 wt.% in this article. During the course of nanocomposite preparation, firstly the nanoparticles and polymer matrix were added into the rotating speed rheometer. Then a percentage of 1010 antioxidant was added into the mixture. After that, the mixture was stirred in 150 °C and the stirring rate of screw was 40 revolutions per minute (RPM). Finally, the nano-MMT/LDPE and nano-ZnO/LDPE composites preparation was completed. The information of different tested samples were shown in Table 1.

**Table 1.** Different tested samples.

| Sample | LDPE (wt.%) | Nanoparticles (wt.%) | Preparation Method |
|---|---|---|---|
| LDPE | 100 | 0 | Melt blending |
| MMT/LDPE | 97 | 3 (MMT) | Melt intercalation |
| ZnO/LDPE | 97 | 3 (ZnO) | Melt blending |

The prepared nanocomposites were laid in a press vulcanizer, in which the temperature was 150 °C, and the pressure was 14.5 MPa. After 15 min pressing, the samples were prepared, of which the thicknesses were 100, and 300 mm, respectively. In order to eliminate the history effect of materials, these samples must be pretreated. Finally, these samples were laid in the vacuum drying oven of which the temperature was 80 °C. After 24 h, these samples could be used for the breakdown field strength test and space charge test.

## 3. Results and Discussion

In order to explore the effect of different morphologies' nanoparticles doping on polymer dielectric properties, the 3 wt.% nano-MMT and nano-ZnO composites were used for test. The compatibility of nanoparticles with polymer matrix was studied and the traps distribution in composite interfacial structure was explored by FTIR and TSC (TSC-650, Toyoseiki, Tokyo, Japan). Meanwhile, the crystalline properties of composites were explored by DSC and PLM (59XB, Shang Guang Technology Co., Ltd., Shanghai, China). Besides, the different samples were dealt with by the breakdown field strength test and the space charge test. Combining the macroscopic experimental research and microcosmic test, the dielectric properties change mechanism of nano-MMT/LDPE and nano-ZnO/LDPE composite was analyzed further.

### 3.1. Chemical Structure Test and Analysis

The samples of N-MMT and O-MMT were tested by X-ray diffraction (XRD, SuperNova, Bruker, Berlin, Germany), the characterization results were shown in Figure 4.

From Figure 4, the θ in the peak of O-MMT was decreased compared with N-MMT, then the interlamellar spacing of O-MMT increased significantly. The diffraction peak of N-MMT and O-MMT appeared in $2\theta = 7°$ and $2\theta = 3.7°$, respectively. According to the Bragg equation $2d\sin\theta = n\lambda$, the interlamellar spacing of O-MMT was 1.91 times larger than which of N-MMT. The XRD results showed that the interlamellar spacing of MMT was expanded after surface modification. The monomer and molecules of polymer was inserted into the interlamellar spacing of MMT easily. Besides, the nano-MMT particles were uniformly dispersed in composites, which reduced the agglomeration effectively. In the range of 3°–10°, there were no apparent peaks of nano-MMT/LDPE composites in XRD patterns. It illustrated the fact that the nano-MMT had separated from the polyethylene matrix.

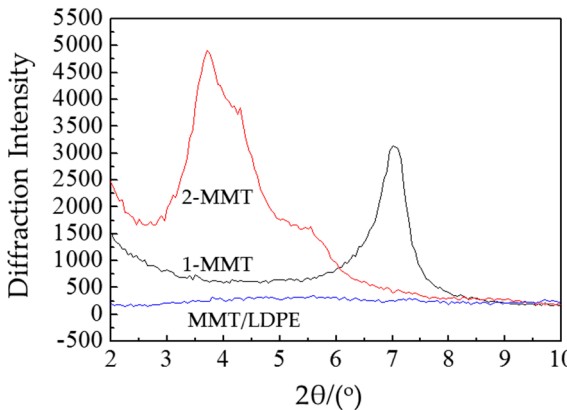

**Figure 4.** XRD patterns of MMT and methylcyclopentadienyl manganese tricarbonyl/low density polyethylene (MMT/LDPE).

The structure of nano-MMT and nano-ZnO before and after surface modification was studied qualitatively by FTIR. The structure and functional group quantity of nanoparticles could be obtained based on different the positions and intensity of absorption peaks. The FTIR model was EQUINOX 55 [25]. Firstly, the KBr, of which wavenumber range was 4000–500 cm$^{-1}$, was used for background material. Then, these particles were crushed and mixed with nano-MMT and nano-ZnO powder, respectively. After that, these samples were rammed by tablet press machine. The experimental results of FTIR were shown in Figure 5.

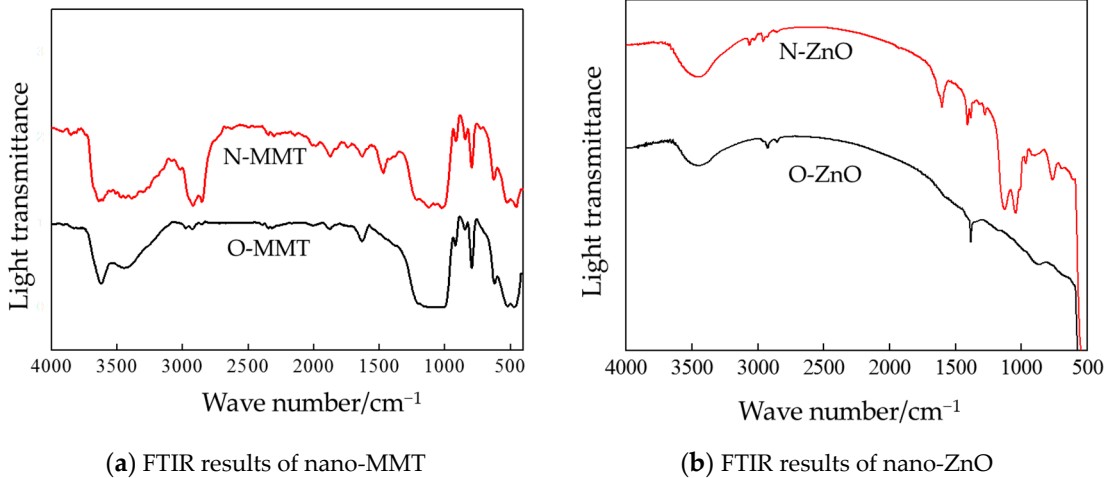

(**a**) FTIR results of nano-MMT         (**b**) FTIR results of nano-ZnO

**Figure 5.** The Fourier-transform infrared spectroscopy (FTIR) results of different nanoparticles before and after surface modification.

From Figure 5a, the H–O–H stretching vibration absorption appeared at around 3624–3640 cm$^{-1}$ and the H–O–H bending vibration absorption appeared at around 1480–1630 cm$^{-1}$, which illustrated that the water molecules existed in nano-MMT layers, and crystal water existed in a lattice [26]. Being compared with N-MMT, the vibration peak value in FTIR of O-MMT decreased, which illustrated that the water molecules and crystal water had been partialluy removed. From FTIR of O-MMT, the –CH$_2$ symmetrical and asymmetrical stretching vibration absorption appeared at around 2850–3000 cm$^{-1}$, which illustrated that the quaternary ammonium salt cationic was inserted into the interlayer of nano-MMT. From the absorption peaks at around 1010–910 cm$^{-1}$, the layered silicate structure of N-MMT and O-MMT had never altered, which illustrated that the chemical bonding between octadecyl trimethyl ammonium chloride and nano-MMT did not happen. It was just the physical adsorption, which changed the surface of nano-MMT from hydrophilic to lipophilic.

From Figure 5b, the hydroxyl (OH–) quantity in O-ZnO was significantly reduced, which was in agreement with the experimental result of chemical titration. The new absorption peaks appeared at around 1200 cm$^{-1}$, which was caused by the stretching vibration of Zn–O–Si. It illustrated that the silanes were transferred to the surface of nano-ZnO particles. So there were plenty of organic groups in the surface of nanoparticles, and the nanoparticles were combined with polymer matrix easily.

### 3.2. Crystalline Structure Test and Analysis

In order to explore the crystallization of LDPE and nanocomposites, the DSC was used to acquire the parameters during the crystallization. This experiment was carried out under the condition of nitrogen protection. The flow rate of nitrogen was set at 150 mL/min, and the rate of heating and cooling was 10 °C/min. The crystallinity of polymer was set to $X_c$, which was calculated by Formula (1):

$$X_c = \frac{\Delta H_m}{(1-w)H_0} \times 100\% \tag{1}$$

where $\Delta H_m$ was the melting enthalpy of material, $H_0$ was the melting enthalpy of material after crystallization, and $w$ was the mass fraction of nanoparticles doping.

Figure 6 was the DSC curve which showed the temperature variation in different samples. The isothermal crystallization parameters and melting parameters of LDPE, nano-MMT/LDPE and nano-ZnO/LDPE were shown in Table 2. Among them, $T_m$ was related to spherulite size of composites. The greater the $T_m$ was, the greater the spherulite size of those composites would be. $T_c$ was the peak temperature of crystallization. It was the temperature which prompted the materials crystallization at top speed. The greater the $T_c$ was, the higher the crystallization temperature of the composites would be.

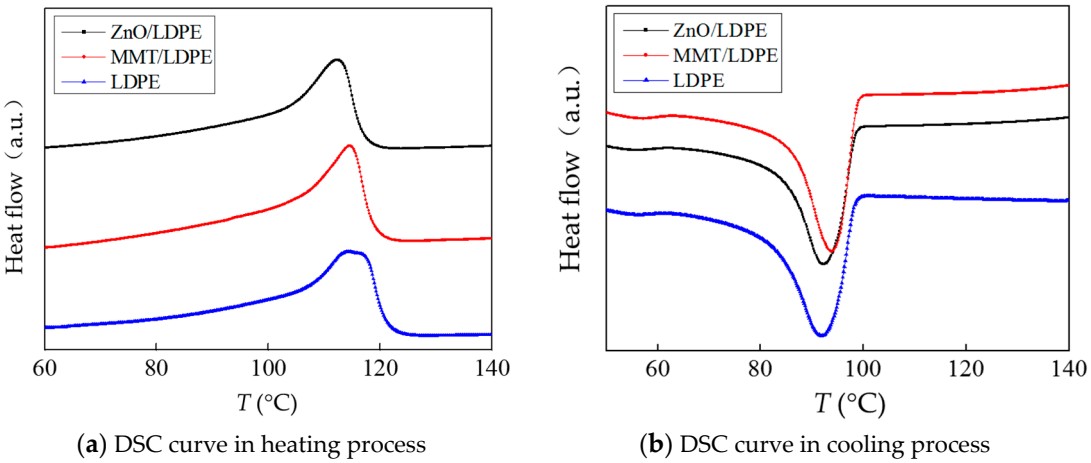

(**a**) DSC curve in heating process　　　　　　(**b**) DSC curve in cooling process

**Figure 6.** Heating and cooling curve of low density polyethylene (LDPE), MMT/LDPE and ZnO/LDPE samples.

**Table 2.** Isothermal crystallization and melting parameters of ZnO/LDPE, MMT/LDPE and LDPE.

| Sample | $T_m/°C$ | $T_c/°C$ | $\Delta T_c/°C$ | $X_c/\%$ |
|---|---|---|---|---|
| LDPE | 117.7 | 92.3 | 17.5 | 36.71 |
| MMT/LDPE | 115.2 | 95.0 | 13.1 | 38.72 |
| ZnO/LDPE | 114.5 | 96.8 | 13.2 | 39.20 |

$T_m$—The melting temperature; $T_c$—The crystallization temperature; $\Delta T_c$—The width of Exothermic crystallization peak; $X_c$—crystallinity.

Combining the result of Figure 6 and Table 2, $T_m$ of nanocomposites was lower than that of pure LDPE. It illustrated that the spherulite size of nanocomposites was smaller [27]. $T_c$ of nanocomposites

was higher than that of pure LDPE. It illustrated that the nanocomposites crystallized at higher temperature. Besides, $\Delta T_c$ of nanocomposites was lower than of pure LDPE. It illustrated that the crystallization speed of nanocomposites was faster. By comparing the results of data analysis, the crystallization speed of nano-ZnO/LDPE was the fastest. Furthermore, the crystallinity of nano-ZnO/LDPE was the highest and the crystalline structure was the most comprehensive.

In order to explore the crystalline morphology further, the different samples were observed by PLM. Firstly, each sample's surface was corroded by the mixture of 5% potassium permanganate and concentrated sulfuric acid for 5 h. In order to corrode the surface of the samples uniformly, the solution was stirred every half an hour during the corrosion process. After that, these samples were removed from the solution and washed by water. Then the ultrasonic cleaner was used to clean them for 15 min. Finally, the crystalline morphology of different samples was obtained by reflected light function of LeicaDM2500 microscope (Leica Microsystems GmbH, Wetzlar, Germany). The crystalline morphology of pure LDPE, nano-MMT/LDPE and nano-ZnO/LDPE were shown in Figure 7.

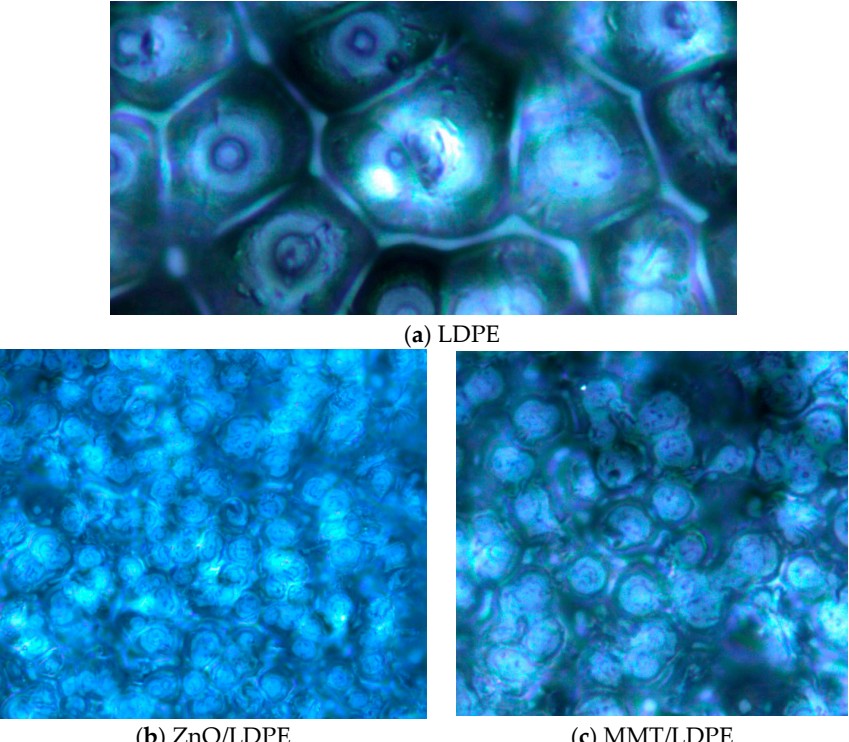

(**a**) LDPE

(**b**) ZnO/LDPE　　　　　　　　(**c**) MMT/LDPE

**Figure 7.** The PLM pictures of different samples.

From Figure 7, the crystal size order of different samples was as follows: LDPE > nano-MMT/LDPE > nano-ZnO/LDPE. The crystal size of pure LDPE, nano-MMT/LDPE and nano-ZnO/LDPE were 45–50, 11–16 and 8–11 μm, respectively. Besides, the amorphous regions size order of different samples was as follows: LDPE > nano-MMT/LDPE > nano-ZnO/LDPE.

In order to characterize the distribution of nano-MMT and nano-ZnO particles in LDPE further, these samples were tested by SEM (Sigma 500, Opton, Beijing, China). Firstly, these samples were processed by brittle segment in liquid nitrogen. Then the fractured surface was sprayed with carbon, from which the micro-structure could be clearly observed. Finally, the dispersion and agglomeration of nanoparticles in polymer would be explored further. SEM results were shown in Figure 8.

From Figure 8b, most of the nano-MMT particles showed a good dispersion in composites. But in some regions, the agglomeration appeared significantly, which illustrated that the interface effect between these nanoparticles and the polymer was relatively weak. From Figure 8c, the dimension of inorganic phase was smaller in nano-ZnO/LDPE. Besides, the nanoparticles dispersed uniformly and

the interface effect appeared clearly. It illustrated that the crystalline morphology of the nano-ZnO composite was more complete.

Combining the result of DSC, PLM and SEM, the cell dimension in LDPE was large and the grains were loosely arranged. Besides, the melting peak and crystallinity were low. After nanoparticles doping, the crystal cell dimension decreased and the grains were closely arranged. Meanwhile, the melting peak and crystallinity increased. The reasons were chiefly as follows: First, there were several elements with favorable thermal conductivity in nano-MMT and nano-ZnO, such as Zn, Si and Al, which transferred heat quickly. It would lead to a rise in the melting peak of this nanocomposite. The second, in the process of cooling crystallization, some nanoparticles acted as a nucleating agent, which led to the close crystalline structure. The other nanoparticles would prevent the crystal cell from increasing, which led to a decrease in crystal size of the nanocomposite. These conclusions also provided a theoretical basis for the follow-up of breakdown experiment.

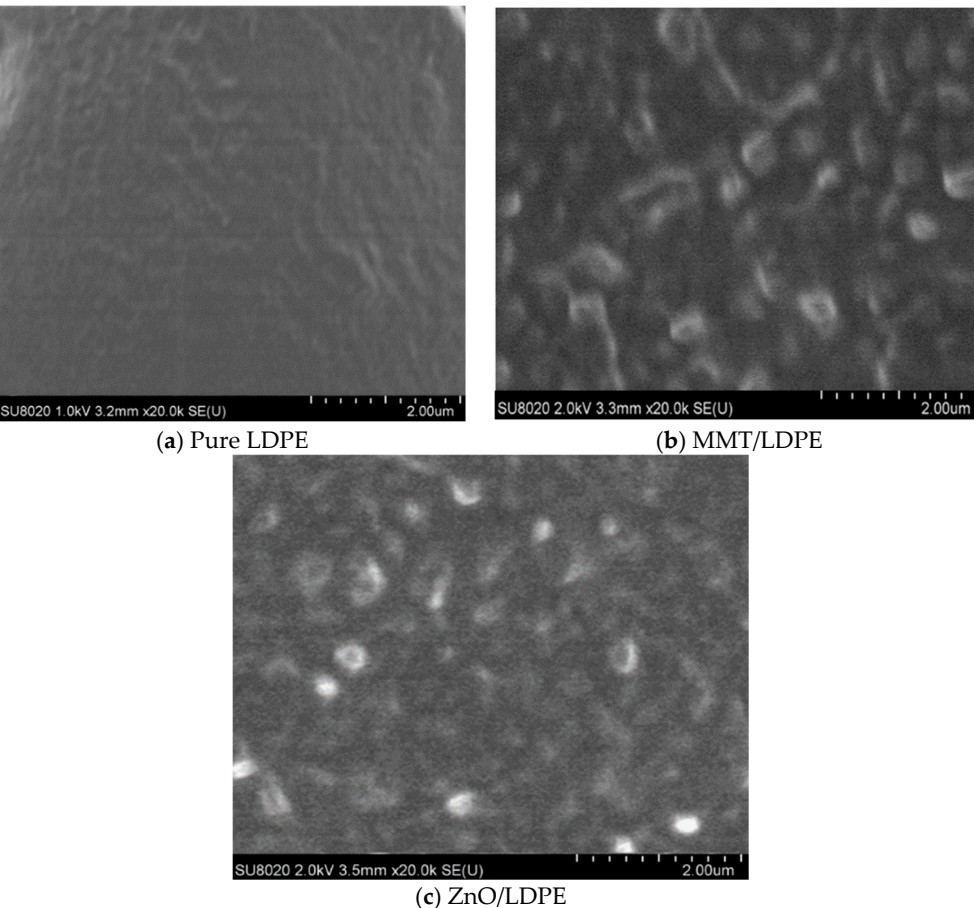

(**a**) Pure LDPE       (**b**) MMT/LDPE

(**c**) ZnO/LDPE

**Figure 8.** The SEM pictures of samples.

*3.3. Breakdown Characteristic Test and Analysis*

In this article, the frequency alternating current system was used to test the breakdown field strength of these different samples. This system boosted at a speed of 1 kV/s until the samples were broken down. The voltage value of breakdown point $U$ was recorded and the thickness of samples in the breakdown point $d$ was measured. After that, the breakdown strength of samples $E$ was calculated by formula $E = U/d$. Each sample was tested for 30 breakdown points, of which data was analyzed by MINITAB 2016a (Version).

The shape parameter β and breakdown field strength $E_0$ under Weibull distribution were obtained. Finally, the Weibull distribution curve of different samples were drawn. β reflected the

dispersion of breakdown voltage in samples. The larger value of β indicated the better dispersion of nanoparticles [28].

In order to explore the effect of different temperatures on the breakdown property of samples, the experimental system was laid in the oven. The experimental temperature was set at 30, 40, 50, 60 and 70 °C, respectively. To prevent the surface discharge, theses samples and the electrode system were placed in a beaker with cable oil. The effect of different temperatures on the breakdown characteristic of the three samples were shown in Figure 9.

From Figure 9, the breakdown field strength of three samples increased initially, and then decreased throughout the testing temperature. When the testing temperature was lower than 60 °C, the breakdown field strength of nanocomposite was higher than that of pure LDPE. Among them, nano-ZnO/LDPE was uppermost at 152.6 kV/mm. However, when the testing temperature was 70 °C, the breakdown field strength of nanocomposites decreased more significantly than that of pure LDPE. The cause of this phenomenon was due to the increase in testing temperature. The influence of impurity level was more serious. The band gap in media became narrow and a conductive channel was easy to form. For these reasons, the breakdown strength decreased.

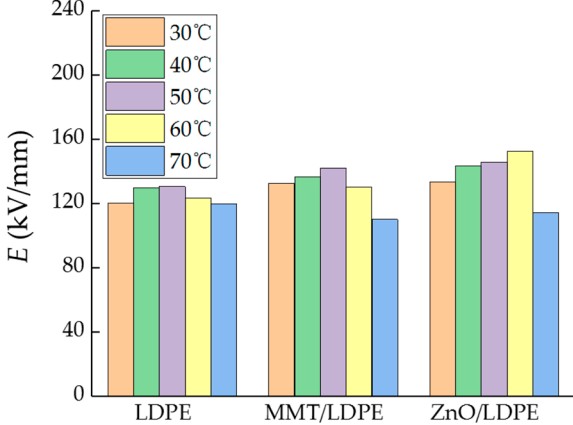

**Figure 9.** The temperature curves of Weibull parameters in different materials.

Under room temperature (25 °C), the test results of the breakdown field strength of these three samples were shown in Figure 10. Meanwhile, the shape parameter and breakdown field strength under Weibull distribution were shown in Table 3.

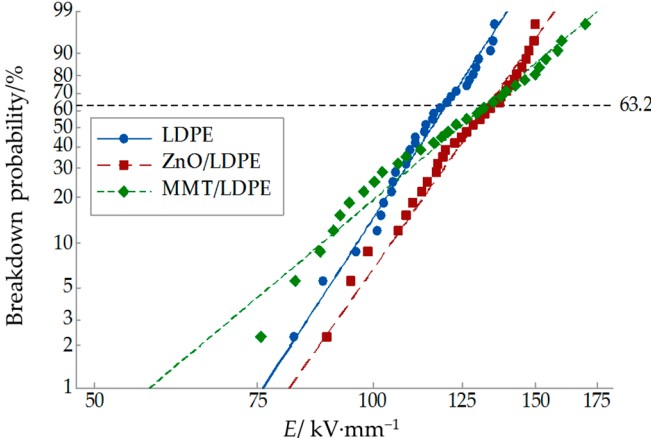

**Figure 10.** The Weibull distribution curve of breakdown field strength in different materials (25 °C).

**Table 3.** Weibull distribution parameters (25 °C).

| Sample | $E/(\text{kV·mm}^{-1})$ | $N$ | β |
|---|---|---|---|
| LDPE | 120.0 | 30 | 10.06 |
| MMT/LDPE | 132.4 | 30 | 5.48 |
| ZnO/LDPE | 133.3 | 30 | 9.24 |

*E*—Breakdown field strength; *N*—The number of breakdown points; β—Shape parameter.

Combining the result of Figure 10 and Table 3, the breakdown field strength of nano-MMT/LDPE and nano-ZnO/LDPE were 10.3% and 11% higher than that of pure LDPE under room temperature (25 °C). The reasons were as follows: The nanoparticles acted as a nucleation agent, which led to the close crystalline structure. Besides, the nanoparticles were dispersed in the matrix homogeneously. The higher barrier would have to be overcome in the electronic formation of conductive paths [29,30]. Being comtransition. So the carriers directional transfer was restrained effectively, which hindered the fobined with the previous analysis of DSC and PLM, nanoparticles doping increased the quantity of crystal cell in composite. The crystal size decreased, and the distribution of crystal cells was close. The amorphous regions decreased, which caused the tortuous conductive path. Besides, most of the carriers transferred in amorphous regions. At the same time, most of nano-MMT and nano-ZnO particles were in the same regions. So the nanoparticles would make a barrier function on electrons. On the one hand, the carriers failed to accumulate enough energy to destroy the molecular chain, on the other hand, the distance of carriers transfer increased. Therefore, the nanoparticles doping improved the breakdown characteristic of materials. Among them, nano-ZnO/LDPE was more obvious.

### 3.4. Space Charge Characteristic Test and Analysis

A large amount of interfacial structure was introduced by nanoparticles doping. In order to explore the trap levels in interfacial structure of nanocomposites further, the different samples were tested by TSC. The thickness of testing samples was 80 μm. In order to eliminate the water and stray charge at the surface of materials, these samples must be pretreated. The samples were laid in the vacuum drying oven, of which the temperature was 80 °C. After the short-circuit treatment for 24 h, the pretreatment was done. Then, the upper and lower surfaces of samples were evaporated into the aluminum electrode with a diameter of 20 mm. During the process of polarization, the 30 kV/mm direct current (DC) electric field was applied to these samples for 1 h in 80 °C. Then the samples were fast cooled below −10 °C by liquid nitrogen. After that, without applied voltage, the samples were short-circuited until the current decreased to 1 pA. Finally, the temperature of samples rose from −10 to 90 °C with the rate of 2 °C/min. In this process, the induced current caused by trapped charges was measured, and the TSC spectra was obtained.

The TSC experimental data of three samples was shown in Figure 11. From this figure, the temperature corresponding to the maximum current value was obtained. The trap depth was calculated by the initial-rise method. Besides, the release of charges was calculated by a curve integral. These data were shown in Table 4.

Combining the result of Figure 11 and Table 4, the shape of spectra was similar in different materials. It illustrated that the formation mechanism of TSC spectra was similar in different materials [31]. The dipolar relaxation did not exist in pure LDPE, so the TSC spectra was produced by trapped charge detrapping [32,33]. Besides, the peak temperature of pure LDPE was 52 °C. According to the analysis of TSC and TL spectra by Ieda, the traps corresponding to the peak temperature around 50 °C were produced by interface defects [34]. After the nanoparticles doping, the peak of TSC spectra shifted toward the high temperature zone. Large amounts of deep traps were introduced. Among them, the trap density of nano-ZnO/LDPE was the greatest. The release of charges in order of different materials was as follows, nano-ZnO/LDPE > nano-MMT/LDPE > LDPE. The greater the release of charges, the more traps would be, and the capability of charges store was better also [35,36].

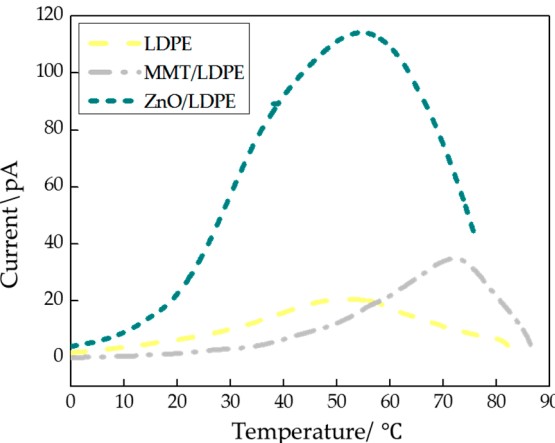

**Figure 11.** TSC spectra of samples.

**Table 4.** Electron trap parameters.

| Name | PE | LDPE/MMT | LDPE/ZnO |
|---|---|---|---|
| Peak temperature/°C | 52 | 72 | 58 |
| Trap depth/V | 0.51 | 0.64 | 0.56 |
| Released of charges/pC | 25,000 | 31,000 | 530,000 |

According to the space charge test, the charges accumulation and release in process of pressurization and short-circuit could be discussed further. Being combined with the previous analysis of TSC, the effect of interface traps structure on space charge was explored. The distribution of space charges in different materials was tested under 30 kV DC electric field and short-circuit respectively. The pressure time was 0.5, 15 and 30 min. In order to observe the distribution of space charges in different materials under pressurization and short-circuit intuitively, the average of charge density in three samples was shown in Figure 12.

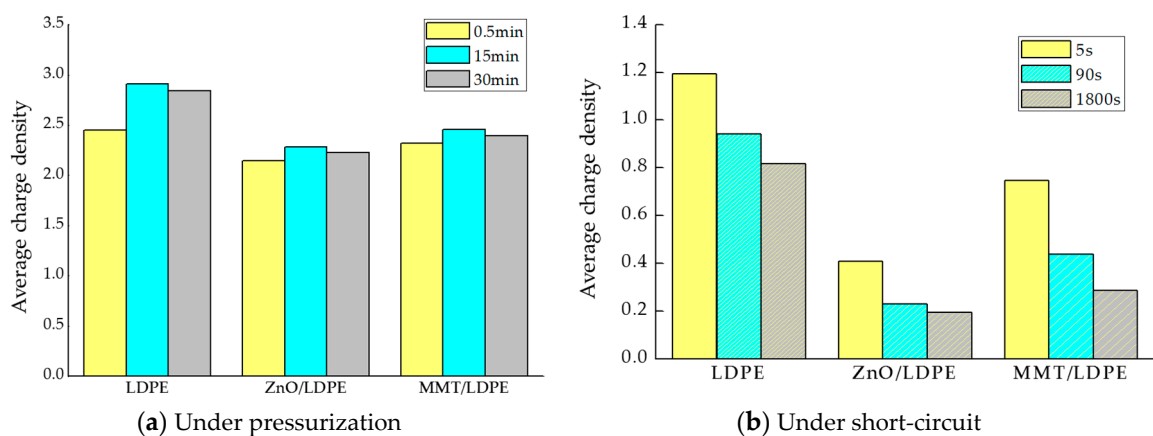

(**a**) Under pressurization　　　　　　　　　　　　(**b**) Under short-circuit

**Figure 12.** Distribution of space charges in different materials under pressurization and short-circuit.

From Figure 12a, the average of charge density in nanocomposites was less than that in pure LDPE under pressurization. Among them, the accumulated charges in nano-ZnO/LDPE was the least. The average of charge density in pure LDPE, nano-MMT/LDPE and nano-ZnO/LDPE at different short-circuit time were shown in Figure 12b. Among them, the charges remnant in nano-ZnO/LDPE was the least.

In summary, a large amount of deep traps were introduced by nanoparticles doping. The carriers were easy to be trapped and hard to get out of the traps. The carrier mobility was improved. Besides,

the free path of the carriers got shorter, and the energy was hard to accumulate. The macromolecule chains in the polymer could not be destroyed. Moreover, these deep traps acted as a recombination center. During the carrier recombination, the photons would be released. If the photons reacted directly on molecule chains of LDPE, the segments would be damaged. But the deep traps were in the bound layer. The structure of bound layer was compact, which was difficult to be destroyed by photons. Besides, the bound layer was very close to the surface of nanoparticles. A large proportion of photons would hit the surface of the nanoparticles, and the energy would be transmitted to nano-MMT and nano-ZnO particles. Because the inorganic materials had good thermal conductivity, the energy accumulation could hardly be formed, and the local failures would rarely happen. In conclusion, the nanoparticles doping could effectively improve the dielectric properties of the polymer.

## 4. Conclusions

- The compatibility of modified nanoparticles with LDPE was excellent. The quantity of crystal cell in nano-MMT/LDPE and nano-ZnO/LDPE increased, the crystal size decreased, and the distribution of crystal cells was close. Being compared with pure LDPE, the crystallization rate of nanocomposite was faster and the crystallinity was higher. Besides, the amorphous regions in nanocomposite decreased, the migration path of carriers was tortuous, and the free path got shorter.

- According to the breakdown field strength test at different temperatures, the breakdown field strength of three samples increased initially, and then decreased with increasing temperature. When the testing temperature was lower than 60 °C, the breakdown field strength of nanocomposite was higher than that of pure LDPE. Under room temperature (25 °C), the breakdown field strength of nano-MMT/LDPE and nano-ZnO/LDPE increased by 10.3% and 11.1% respectively, compared to that of pure LDPE.

- According to the result of TSC, the trap density of nanocomposites was great, and many deep traps existed in materials. The carriers were captured effectively. The electric field strength decreased and the dielectric properties of the polymer improved. According to the space charge distribution test, the space charge accumulation in LDPE was restrained by nanoparticles doping. Among them, the space charges in nano-ZnO/LDPE was the least, and the release rate of charges in nano-MMT/LDPE was the fastest under short-circuit.

**Author Contributions:** Conceptualization, G.Y.; Methodology, Y.C. and G.Y.; Formal Analysis, G.Y. and X.Z.; Investigation Y.C.; Resources, X.Z.; Data Curation, Y.C., G.Y.; Writing—Original Draft Preparation, Y.C.; Writing—Review and Editing, G.Y.; Supervision, X.Z.; Project Administraction, Y.C.; Funding Acquisition, X.Z.

**Funding:** This research was funded by National Natural Science Foundation of China which is Investigation on meso-morphology and mechanism inhibiting space charge in polymer/inorganic micro-nano composites based on the coordination effect of micro- and nano-filler.

**Acknowledgments:** Thank X.Z. and Y.C. for their help.

**Conflicts of Interest:** The authors declare no conflicts of interest.

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
