# Peer review of "The Dielectric Properties Improvement of Cable Insulation Layer by Different Morphology Nanoparticles Doping into LDPE"

_coatings, doi:10.3390/coatings9030204_

Round 1
Reviewer 1 Report
Author documented this manuscript with detailed explanation about the dielectric properties improvement of cable insulation layer by different morphology inorganic nanoparticles doping into LDPE and the samples were characterized. However, some of the arguments by author not should be clarified.
1. Author should go through the whole manuscript carefully and fix the typos. For example, page 4, line 107, author mentioned 40RPM. There is no space between 40 and RPM. It should be mentioned 40 RPM. Similarly, 15min, 14.5Mpa, and 300mm etc.
2. Author should indicate (a) and (b) in Figure 4.
3. Author explained in Figure 4, FTIR was used to illustrate that the quaternary ammonium salt cationic was inserted into the layer of nano-MMT and confirmed that the intercalation was happened. The values are not clear what author mentioned in the manuscript. Can author do powder XRD pattern to prove the before and after intercalation of quaternary ammonium salt?
Author Response
Author Response to Reviewer 1:
1. Author should go through the whole manuscript carefully and fix the typos. For example, page 4, line 107, author mentioned 40RPM. There is no space between 40 and RPM. It should be mentioned 40 RPM. Similarly, 15min, 14.5Mpa, and 300mm etc.
The typos has been fixed.
2. Author should indicate (a) and (b) in Figure 4.
They have been indicated.
3. Author explained in Figure 4, FTIR was used to illustrate that the quaternary ammonium salt cationic was inserted into the layer of nano-MMT and confirmed that the intercalation was happened. The values are not clear what author mentioned in the manuscript. Can author do powder XRD pattern to prove the before and after intercalation of quaternary ammonium salt?
The XRD experiment has been added into this article. It illustrated that the nano-MMT had separated from polyethylene matrix. The LDPE was inserted into interlamellar spacing of MMT easily.

Reviewer 2 Report
Yu et al. investigated the dielectric properties of cable insulation layer that were shown to increase by employing LDPE-doped MMT or ZnO. I think that the authors show some good properties including ability to improve the dielectric properties of polymer. The overall concept is OK, but the English grammar is very poor. The authors would be well advised to carefully check over the grammar.
1. Although PLM images are provided, quantitative characterization of the surfaces (i.e., roughness) using AFM would be nice. Or, SEM images of the composites could support the structures of the grains.
2. The authors need to provide a more detailed explanation over the table 2, which discusses Tm and Tc of the composites.
3. Figure 1 that describes experimental equipment should move to SI.
4. There is lots of clumsy grammar in Abstract. And the title is too long.
5. I think that the authors’ explanation for the TSC spectra is reasonable, but does it have any other characterization to directly show a space charge?
Author Response
Author Response to Reviewer 2:
1. Although PLM images are provided, quantitative characterization of the surfaces (i.e., roughness) using AFM would be nice. Or, SEM images of the composites could support the structures of the grains.
The SEM experiment has been added into this article, from which the dispersion of nanoparticles in polymer could be observed.
2. The authors need to provide a more detailed explanation over the table 2, which discusses Tm and Tc of the composites.
The instructions of Tm and Tc were added into this article.
3. Figure 1 that describes experimental equipment should move to SI.
Figure 1 has been adjusted.
4. There is lots of clumsy grammar in Abstract. And the title is too long.
The English language in this article has been improved, and the title was simplified.
5. I think that the authors’ explanation for the TSC spectra is reasonable, but does it have any other characterization to directly show a space charge?
The traps density and depth in samples was characterized by TSC. Besides, the space charge distribution experiment was carried out in this article. The experimental result was displayed in figure 12 directly. Meanwhile, to avoid any confusion, the experimental images which described the same phenomenon repeatedly were deleted.

Round 2
Reviewer 1 Report
Author has included XRD pattern and explained in detail about N-MMT and O-MMT pattern. However, author should go through the manuscript carefully and fix some typos.
For example, In page 2, line 68, author mentioned 30nm but it should be 30 nm. Similarly, in the same line 68, 80mol/100g. Similarly, in line 112, author mentioned 100mm and 300mm. Overall the manuscript is good.
Reviewer 2 Report
The authors have satisfied my concerns. It is an interesting system with good properties.